# Molecular Protein and Expression Profile in the Primary Tumors of Clear Cell Renal Carcinoma and Metastases

**DOI:** 10.3390/cells9071680

**Published:** 2020-07-13

**Authors:** Liudmila V. Spirina, Zahar A. Yurmazov, Alexey K. Gorbunov, Evgeny A. Usynin, Nadezhda A. Lushnikova, Irina V. Kovaleva

**Affiliations:** 1Cancer Research Institute, Tomsk National Research Medical Center of the Russian Academy of Sciences, Tomsk 634050, Russia; pzahar@sibmail.com (Z.A.Y.); goorbunovak@yandex.ru (A.K.G.); gusi@list.ru (E.A.U.); nadejdaland@gmail.com (N.A.L.); 2Siberian State Medical University, Tomsk 634050, Russia; irina.kovalyova.kovaleva@mail.ru

**Keywords:** kidney cancers, PTEN, transcription factors, growth factors, AKT/mTOR components, primary tumors, metastatic sites

## Abstract

Metastasis involves the spread of cancer cells from the primary tumor to surrounding tissues and distant organs and is the primary cause of cancer morbidity and mortality. The aim of the study was the determination of change in molecular factors expression in primary kidney cancers (ccRCC) and metastatic sites. In total, 62 patients with RCC were enrolled in the study. The mRNA levels of molecular markers were studied by real-time PCR, and the content of the studied parameters was determined by Western blotting and ELISA. The features in the intracellular signal metabolites in the series of normal renal parenchyma, tumor tissue of localized, disseminated kidney cancer and metastatic tissue were studied. A decrease in some indicators in the tissue of the metastatic lesion was noted. Protein products of transcription factors HIF-1, CAIX, PTEN and activated AKT kinase, as well as expression of the VEGFR2 receptor and m-TOR protein kinase were revealed to be reduced in the metastatic sites. In addition, some indicators increased in metastasis: the protein levels of NF-κB p 50, NF-κB p 65, HIF-2, VEGF, VEGFR2, m-TOR and mRNA of HIF-1, CAIX, PTEN and PDK. There were indicators with multidirectional changes. HIF-1, CAIX, PTEN, VEGFR2 and m-TOR mRNA: VEGFR2, m-TOR, HIF-1, CAIX, PTEN and PDK had an opposite change in protein content and mRNA level. PTEN loss resulted in the downstream activation of AKT/mTOR signaling in secondary cancer lesions and determined the overall ccRCC patient’s survival. The AKT/mTOR signaling cascade activation was found in the primary kidney tumors. The PTEN content and mRNA level were correlated with total AKT, GSK-3β, the 70S 6 kinases and AKT expression.

## 1. Introduction

Metastasis involves the spread of cancer cells from the primary tumor to surrounding tissues and to distant organs and is the primary cause of cancer morbidity and mortality [1]. The metastatic cascade includes the cancer cells detaching from the primary tumor, extravasation at distant capillary beds and invasion of distant organs [2]. Currently, several hypotheses have been advanced to explain the origin of cancer metastasis. These involve an epithelial–mesenchymal transition, an accumulation of mutations in stem cells, a macrophage facilitation process and a macrophage origin involving either transformation or fusion hybridization with neoplastic cells. Metastases are responsible for the vast majority of cancer-related deaths. Although tumor cells can become invasive early during cancer progression, metastases formation typically occurs as a late event [2,3].

Metastasis is a process consisting of cells spreading from the primary site of cancer to distant parts of the body. There is some evidence that primary clear cell renal cell carcinoma (ccRCC) and metastases of RCC exhibit molecular differences that may affect the biological characteristics of the tumor [1]. Metastatic cells also establish a microenvironment that facilitates angiogenesis and proliferation, resulting in macroscopic, malignant secondary tumors. Although systemic metastasis is responsible for about 90% of cancer deaths, most research on cancer does not involve metastasis in the in vivo state [3].

The identification of predictive factors of response is critical for the development and appropriate use of anti-cancer agents. The evaluation of biomarkers is usually performed by analyzing the primary tumor tissues, but this approach does not account for potential discrepancies between primary tumor and secondary lesions [4].

We attempted to summarize studies connected with differences between primary ccRCC and its metastases and their influence on the biological characteristics of renal cancer. Overall, metastases were more similar to each other and often differed from the primary tumor [5]. However, most of the distant metastases had heterogeneous molecular profiles. Molecular profiles of metastases may vary from primary cancer and each other, probably resulting in a different response to therapy. Although metastases and primary tumors share standard histological features, this study highlighted chromosomal differences specific to metastases, which could be involved in ccRCC metastatic evolution [6]. These data support the theory that ccRCC primary tumors and metastases encompass a uniform distribution of common genomic alterations tested by next-generation sequencing targeted panels. This study did not address variability between matched primary tumors and metastases or the change in genomic modifications over time and after sequential systemic therapies [7].

Von Hippel–Lindau protein (VHL) deficiency is a dominant feature in ccRCC [8]. VHL substrates have been identified, including members of the hypoxia-inducible (HIF) transcription factor family. They translocate to the nucleus, activate 100–200 genes that contain hypoxia-response elements, and promote adaptation to hypoxia through the transcription of VEGF and CAIX genes [9].

PI3K/AKT/mTOR activation is a result of growth factors signal transduction. It is one of the universal signaling pathways characteristic of most cells, the central components of which are the enzymes phosphoinositide-3-kinase (PI3K), AKT and m-TOR kinases [10]. In the presence of oxygen, HIFα is hydroxylated, bound to VHL and then degraded by the proteasome [9]. VHL lack results in the accumulation of HIFs and AKT/mTOR overactivation [8].

PTEN is a key suppressor of AKT/mTOR signaling cascade. Its mutations and deletion within primary tumors have been associated with an increased risk of metastasis, and new targeting of PTEN may prevent metastasis [11]. Decreased PTEN has been associated with poorer survival outcomes of patients with kidney cancer. PTEN acts as a tumor suppressor in tumorigeneses and progression in kidney cancer [12], influencing the transcriptional and protein levels of molecular factors.

Alterations in molecular factors expression characterize distant metastases of different metastatic sites. They could be a helpful tool for individual follow-up prediction and personalized therapy selection [13]. A more robust pS6 expression and more frequent overexpression in metastases were shown than in corresponding primary renal cell cancers. In approximately one-third of the cases, pS6 overexpression was found exclusively in metastases, which is interesting considering the association between high pS6 expression and sensitivity to mTOR inhibitor therapy [14,15,16]. Endothelin receptor type B, phos-S6 and CD44 are variously expressed in primary ccRCCs and their metastases and have effect on clinical outcome [17]. Our understanding of this spread is limited and molecular mechanisms causing particular characteristics of metastasis are still unknown.

This study aimed to determine the change in molecular factors protein expression and mRNA level in primary kidney cancers and metastatic sites and to find the associations with PTEN.

## 2. Materials and Methods

In total, 62 patients with ccRCC were enrolled in the study (22 females and 40 males). The median age of the patients was 57.0 ± 9.2 years. The patients were admitted to and nephrectomized at the Cancer Research Institute, Tomsk National Research Center, Russian Academy of Medical Sciences, Tomsk, Russian Federation. The patients underwent a physical examination, chest radiography and computer tomography (CT) of the abdomen. When vena cava tumor thrombus invasion was suspected, cavography or magnetic resonance imaging (MRI) was performed. Patients with skeletal-associated pain or elevated serum alkaline phosphatase were assessed with bone scintigraphy. The patients were followed-up according to a program including regular clinical and radiological examinations. The treatment for kidney cancer depends on the size of the cancer and whether it has spread to other parts of the body. Diagnosis was verified on the basis of biopsy results.

Localized ccRCC (T1-3N0M0) was diagnosed in 18 patients and metastatic ccRCC (T2-4N0-1M1) in 44 patients. All patients with localized RCC underwent surgery (partial nephrectomy or simple nephrectomy) and then followed-up according to a program including regular clinical and radiological examinations. Patients with metastatic ccRCC received two cycles of preoperative targeted therapy with pazopanib at a dose of 800 mg daily, for two months. Tumor response to targeted therapy was evaluated according to RECIST criteria. All patients underwent radical nephrectomy.

The Local Committee for Medical Ethics approved the study, and all patients provided written informed consent. Biopsy samples of normal renal parenchyma, tumor tissues of localized and disseminated cancers and metastatic tissues were used for investigation. Specimens were reviewed separately by two independent pathologists. 

### 2.1. RNA Extraction

The postoperative tumor samples were incubated in RNAlater solution (Ambion, Austin, TX, USA) for 24 h at +4 °C and then stored at −80 °C. Total RNA was extracted using RNeasy Mini Kit (Qiagen, Hilden, Germany).

RT-qPCR. PCR was conducted in 25-μL reaction volumes containing 12.5 μL BioMaster HS-qPCR SYBR Blue (2×) (“Biolabmix”, Novosibirsk, Russia) and 300 nM of each primers (Table 1).

A pre-incubation at 95 °C for 10 min was performed to activate the Hot Start DNA polymerase and denature DNA, which was followed by 45 amplification cycles of 95 °C denaturation for 10 s and 60 °C annealing for 20 s (iCycler iQ™, BioRad, Hercules, CA, USA).

The fold changes were calculated by ΔΔCt method (the total ΔΔCt = fold of cancerous/normal tissue gene level), using normal tissue. A ratio of specific mRNA/GADPH (GADPH as a respective control) amplification was then calculated. 

VEGF, VEGFR2, CAIX, HIF-1α, HIF-2, NF-κB p50 and NF-κB p65 determination: HIF-1α and NF-κΒ (p50 and p65) expressions were measured using Caymanchem ELISA kits (Ann Arbor, MI, USA) in Anthos 2020 ELISA-microplate reader (Biochrom, Cambridge, UK). Nuclear extracts were prepared and purified according to manufacturer’s instructions. CAIX and HIF-2α determination was performed using Cusabio Elisa kits (Wuhan, China).

### 2.2. Determination of Expression Levels of AKT/M-TOR Signaling Pathway Components

Electrophoresis SDS-PAGE (Laemmli) was used. The protein was transferred to 0.2 μm pore-sized PVDF membrane (GE Healthcare, Amersham, UK), either at 150 mA or 100 V for 1 h by using a Bio-Rad Mini Trans- Blot electrophoresis cell. The membrane was incubated in a 1:2500 dilution of monoclonal mouse anti-human phospho-PTEN (Ser380), AKT (pan), phospho -AKT (T308), phospho-GSK-3-beta (Ser9), phospho-PDK1 (Ser241), phospho-c-Raf (Ser259), m-TOR, phospho-mTOR (Ser2448), phospho-p70 S6 (Ser371) and phospho-4E-BP1 (Thr37/46) (Cell Signaling, Beverly, MA, USA) at 4 °C overnight.

PVDF samples were incubated in Amersham ECL Western blotting detection analysis system (Amersham, UK). The results were standardized using the beta-actin expression in a sample and were expressed in percentages to the protein content in non-transformed tissues. The level of protein in normal kidney parenchyma was indicated as 100%.

### 2.3. Statistical Analysis

Statistical analysis was performed using SPSS 19.0 software. Data were expressed as median and ranges. Mann–Whitney test was used for comparing differences in mean values. nonparametric one-way ANOVA on ranks was carried out for testing whether samples originate from the same distribution, which is used for comparing two or more independent samples of equal or different sample sizes. Nonparametric correlation analysis was performed, and the Spearmen coefficient was calculated.

## 3. Results

The changes in the studied markers in the normal renal parenchyma, localized tumor tissue, disseminated kidney cancer and metastatic tissue were studied (Table 2). In a tumor of localized kidney cancer, an increase in the level of transcription factors was observed compared with normal renal parenchyma (HIF-1 by 4.8 times; HIF-2 by 1.5 times; NF-kB p50 by 1.8 times; and NF-kB p65 by 1.6 times). The increase in the transcription factors expression was accompanied by the high values of the growth factor VEGF, whose level was 5.2 times higher than in the normal kidney parenchyma (Table 2).

The most aggressive course is the disseminated form of the disease, in the tissue of the primary tumor of which there is an increase in the content of NF-kB p50 (1.6 times), NF-kB p65 (1.4 times), HIF-1 (5.5 times) and HIF-2 (1.3 times) compared with normal renal parenchyma. The corresponding activation of transcription factors leads to an increase in the content of VEGF, VEGFR2 and phospho-mTOR by 6.7, 1.6 and 1.59 times, respectively, compared with unchanged tissue. The CAIX level in the tumor tissue was equal to that in normal tissue and decreased by 1.76 times compared with the same indicator in the primary tumor of localized cancer.

We detected no significant changes in the content of NF-kB p50 in metastatic tissue. At the same time, the NF-kB p65 level underwent significant changes: it decreased by 2.0 times in metastases compared with the primary tumor and corresponded to the level of this indicator in normal renal parenchyma. A decrease in the content of transcription factor HIF-1 by three times in metastatic foci as compared with tumor tissue was also detected. At the same time, its level was increased by 1.7 times compared with normal renal parenchyma. A progressive increase in the content of nuclear factor HIF-2 occurred in metastatic tissue. It was increased by 5.0 times compared with normal tissue and 3.4 times compared with the tissue of localized and disseminated kidney cancers.

A change in the HIF-1 expression was noted in metastatic sites compared with the primary tumor of localized and disseminated kidney cancers. An increase in the mRNA level of this indicator of 13 and 40.3 times, respectively, was noted. Therefore, the growth of the protein product was combined with the increase of the corresponding mRNA, which is probably crucial for the processes of tumor progression.

In turn, the content of growth factor VEGF in tumor metastases remained high, which also corresponded to the level of the indicator in the primary tumor. A significant increase in the level of VEGFR2 by a factor of 2.3 was noted in metastatic sites compared with tumor tissue of disseminated kidney cancer (Table 3). In contrast to the molecular markers described above, the level of CAIX in metastatic tissues was decreased by 3.9, 4.7 and 3.0 times, respectively, compared with normal tissue and the tissue of the primary tumor of localized and disseminated cancers. This confirmed the above-described trends in the expression of transcription factor HIF-1 in metastatic tissue.

We found an opposite change in the VEGFR2 and CAIX expression in metastatic sites. A 1.4-fold decrease in the level of VEGFR2 mRNA and a 49.5-fold increase in the CAIX mRNA level were noted in comparison with the primary tumors. 

Table 4 presents data on the expression of components of the AKT/mTOR signaling pathway in the tissue of localized, locally advanced and disseminated kidney cancer, as well as in metastatic sites. It should be noted that the level of activated AKT, phospho-AKT (T308) and phospho-AKT (S473) was decreased by 1.4 and 1.38 times, respectively, in patients with disseminated kidney cancer compared with patients without metastases. In metastases, similar trends in the expression of molecular factors were observed. Identified changes were confirmed using a nonparametric analysis of variance (Kruskal–Wallis test). It is worth noting that a difference in the expression of PDK kinase in the primary tumor of localized cancer, disseminated cancer and metastases was also detected. The most pronounced decrease in this indicator was noted in the tumor tissue in the disseminated kidney cancers (9.0 times) compared with the primary tumor in the localized one.

A 1.56-fold increase in the content of phospho-m-TOR in metastatic sites compared with the tissue of localized kidney cancer was shown to be combined with a 9.6-fold decrease in mRNA levels. At the same time, significant changes in the expression of phospho-p70 S6 kinase and phospho-4E-BP1 were not detected.

The study revealed a significant decrease in the level of phospho-PTEN by 1.8 times in patients with a disseminated form of the disease compared with patients with localized one (Table 3). Moreover, the level of PTEN mRNA was increased in metastatic tissue by 2.5 and 160.7 times, respectively, compared with the primary tumor of localized and disseminated cancers.

According to the correlation analysis, associations between the protein level of PTEN with a total AKT level in the primary tumors (r = 0.54; p = 0.0005) were revealed (Figure 1). PTEN mRNA level was correlated with the GSK-3β, 70S 6 kinase and AKT expression (r1 = 0.34; p1 = 0.018; r2 = 0.40; p2 = 0.004; r3 = 0.33; p3 = 0.02).

## 4. Discussion

Alterations in molecular factors expression could be a helpful tool for individual follow-up prediction and personalized therapy selection [13]. We assumed that the more aggressive behavior of tumor cells in metastases is associated with a high level of nuclear factor HIF-2. Thus, the content of changes in molecular indicators increases as the tumor spreads in the body, which is most pronounced in patients with a disseminated form of the disease. Overexpression of the transcription factor HIF-2 with a decrease in the level of HIF-1 leads to the formation of the metastatic site. The activation of HIF-2 against hypoxic stress leads to more pronounced changes in the proliferative potential of tumor cells with activation of their active distribution [18].

PI3K/AKT/mTOR is one of the universal signaling pathways characteristic of most cells, the central components of which are the enzymes phosphoinositide-3-kinase (PI3K), AKT and m-TOR kinases [19]. This intracellular cascade is critical in the life of the cell, determining its growth, proliferation and apoptosis [20,21,22]. Our study aimed at investigating the features of the activation of the complex of intracellular signal metabolites in the series of normal renal parenchyma, tumor tissue of localized, disseminated kidney cancer and metastatic tissues.

We showed the changes in protein and mRNA profiles in primary cancers and metastatic sites. It is known the primary tumors and their corresponding metastases differ from each other in gene, mRNAs and proteins profiles [6]. An increase in endothelin receptor type B, phospho-S6 and CD44 expression and more frequent overexpression in metastases than in corresponding primary renal cell cancers were shown [13,17,18]. 

Figure 2 shows the predefined changes in the profile of proteins and mRNA of molecular factors that play an essential role in the processes of tumor progression. A decrease in some indicators in the tissue of the metastatic lesion was noted. Protein products of transcription factors HIF-1, CAIX, PTEN and activated AKT kinase, as well as expression of the VEGFR2 receptor and m-TOR protein kinase were revealed to be reduced in the metastatic sites. In addition, some indicators increased in metastasis: protein levels of NF-κB p 50, NF-κB p 65, HIF-2, VEGF, VEGFR2 and m-TOR and mRNA levels of HIF-1, CAIX, PTEN and PDK. There are indicators with multidirectional changes. HIF-1, CAIX, PTEN, VEGFR2 and m-TOR mRNA: VEGFR2, m-TOR, HIF-1, CAIX, PTEN and PDK had an opposite change in protein content and mRNA level.

The importance of PTEN in the tumor progression of kidney cancer is extremely complex and ambiguous [11,12]. Undoubtedly, its effect on the state of intracellular signaling cascades is crucial in oncogenesis. The study confirmed a progressive decrease in protein levels in metastatic tumors compared with non-metastatic tumors. At the same time, the mRNA level sharply increased against the background of tumor progression. The leading event is probably the PTEN loss and the increase in its functional inability.

Angiogenesis plays a critical role in the growth of cancer and its spreading. To grow beyond a limited size, all solid cancers require a proper vasculature. The angiogenic program in ccRCC is switched on deficiency of VHL protein, resulting in excess of HIF-1 proteins, which recruit vascular cells for the tumor vessel plexus formation [8]. HIF proteins trigger the transcription of several genes that are key mediators of the angiogenic process, such as VEGF and CAIX [9]. The overactivation of the VHL/HIF/VEGF/VEGFRs axis is a trend in ccRCC; this justifies the marked sensitivity of this neoplasm to antiangiogenic agents [23]. The angiogenic factors can be used for prognostics, predicting unfavorable patient’s outcome and varying the cancer administration. The reduction in the expression level of VEGF is a sign of improved quality of life [19]. 

The revealed data on the significant increase in VEGFR2 and m-TOR protein levels and decrease in HIF-1 and PTEN in metastatic sites show the promising approach in the management of disseminated ccRCC. The PTEN deficiency is a key event in the angiogenic imbalance. It results in the switching on the angiogenesis and causes the ccRCC progression [23]. PTEN deficiency contributes to the development and progression of ccRCC. Shorter overall survival in ccRCC patients is correlated with low PTEN expression [12,24].

## 5. Conclusions

PTEN loss results in the downstream activation of AKT/mTOR signaling in secondary cancer lesions and determines the overall ccRCC patient’s survival. The PTEN content and mRNA level are correlated with a total AKT, GSK-3β, the 70S 6 kinases and AKT expression. The data highlight the variety of molecular factors implicated in cancer progression and switching on the metastatic phenotype. Overactivation of angiogenic factors governing HIF-1 leads to the increase in VEGFR2 content in metastatic sites, accompanied by the growth in m-TOR protein level. The AKT signaling cascade activation was found in the primary kidney tumors. NF-κB p 50, κB p 65, HIF-1, HIF-2, VEGF, CAIX, VEGFR2 mTOR, PTEN and AKT are molecular factors characterizing the distant metastases. Biological mechanisms causing particular characteristics of sites of secondary lesions serves as the predictive factors of response to anti-cancer agents. 

## Figures and Tables

**Figure 1 cells-09-01680-f001:**
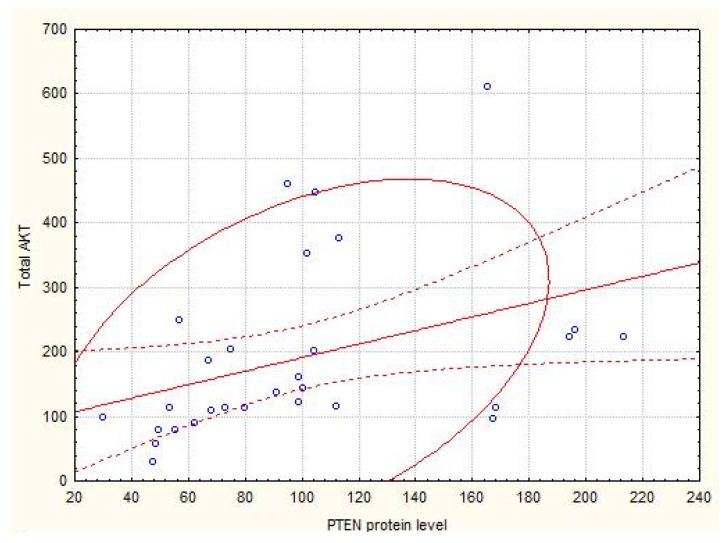
Scatterplot of PTEN and total AKT protein level in kidney cancers. Note: We found the associations between the protein level of PTEN with a total AKT (AKT pan) level in the primary tumors (*r* = 0.54; *p* = 0.0005), indicating the phosphatase influence on downstream of intracellular signaling.

**Figure 2 cells-09-01680-f002:**
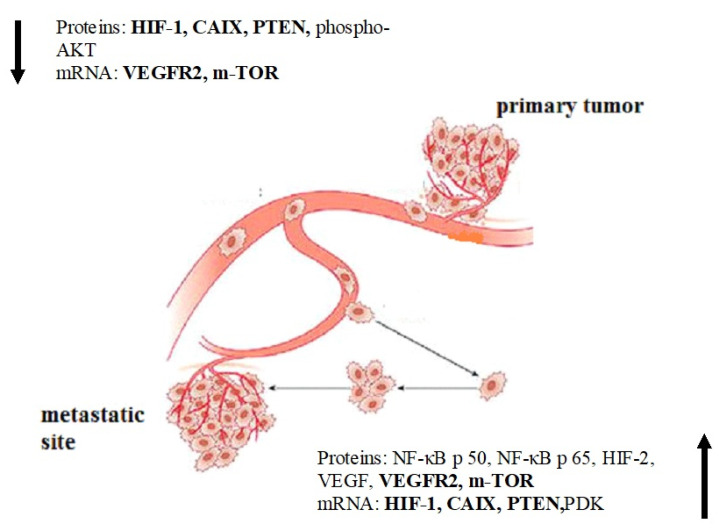
Change in the profile of proteins and mRNAs of molecular factors in the primary tumor of the kidney and its metastatic sites. Note: Arrows indicate changes in the studied factors. We found a decrease in HIF-1, CAIX, PTEN, phospho-AKT content and RNA levels of VEGFR2 and m-TOR in metastases compared to the primary tumors. NF-κB p50, NF-κB p65, HIF-2, VEGF VEGFR2 and m-TOR protein contents and HIF-1, CAIX, PTEN and PDK RNA levels were enhanced in secondary cancer lesions. Indicators with change in opposite direction are highlighted in bold. Proteins level and mRNA alterations show the specific trend in biological features of primary tumors and metastases. Despite the reduction in mRNA levels of VEGFR2 and m-TOR, the protein contents of these molecular factors increased. The HIF-1 and PTEN protein levels showed the opposite trend. The kidney cancer aggressiveness is associated with the overactivation of AKT/mTOR signaling pathway and excess of tyrosine kinase receptors.

**Table 1 cells-09-01680-t001:** The sequence of primers.

Gene	Amplicon	Sequence
*CAIX* NM_001216.2	217 bp	F 5′-GTTGCTGTCTCGCTTGGAA-3′ R 5′-CAGGGTGTCAGAGAGGGTGT-3′
*HIF-1α* NM_001243084.1	188 bp	F 5′- CAAGAACCTACTGCTAATGCCA-3′ R 5′- TTTGGTGAGGCTGTCCGA-3′
*EPAS1* NM_001430.4	265 bp	F 5′- TGGAGTATGAAGAGCAAGCCT-3′ R 5′-GGGAACCTGCTCTTGCTGT-3′
*NFKB1* NM_001165412.1	144 bp	F 5′-CGTGTAAACCAAAGCCCTAAA-3′ R 5′-AACCAAGAAAGGAAGCCAAGT-3′
*RELA* NM_001145138.1	271 bp	F 5′-GGAGCACAGATACCACCAAGA-3′ R 5′-GGGTTGTTGTTGGTCTGGAT-3′
*PTEN* NM_001304717.2	136 bp	F 5′-GGGAATGGAGGGAATGCT-3′ R 5′-CGCAAACAACAAGCAGTGA-3′
*VEGFA* NM_001025366.2	316 bp	F 5′-AGGGCAGAATCATCACGAA-3′ R 5′-TCTTGCTCTATCTTTCTTTGGTCT-3′
*KDR* NM_002253.2	306 bp	F 5′-AACACAGCAGGAATCAGTCA-3′ R 5′-GTGGTGTCTGTGTCATCGGA-3′
*4EBP1* NM_004095.3	244 bp	F 5′- CAGCCCTTTCTCCCTCACT -3′ R 5′- TTCCCAAGCACATCAACCT -3′
*AKT1* NM_001014431.1	181 bp	F 5′- CGAGGACGCCAAGGAGA-3′ R 5′- GTCATCTTGGTCAGGTGGTGT-3′
*C-RAF* NM_002880.3	152 bp	F 5′- TGGTGTGTCCTGCTCCCT-3′ R 5′- ACTGCCTGCTACCTTACTTCCT-3′
*GSK3b* NM_001146156.1	267 bp	F 5′- AGACAAGGACGGCAGCAA-3′ R 5′- CTGGAGTAGAAGAAATAACGCAAT-3′
*70S kinase alpha* NM_001272042.1	244 bp	F 5′- CAGCACAGCAAATCCTCAGA-3′ R 5′- ACACATCTCCCTCTCCACCTT-3′
*m-TOR* NM_004958.3	160 bp	F 5′- CCAAAGGCAACAAGCGAT-3′ R 5′- TTCACCAAACCGTCTCCAA-3′
*PDK1* NM_001278549.1	187 bp	F 5′- TCACCAGGACAGCCAATACA-3′ R 5′- CTCCTCGGTCACTCATCTTCA-3′
*GAPDH* NM_001256799.2	138 bp	F 5′-GGAAGTCAGGTGGAGCGA-3′ R 5′-GCAACAATATCCACTTTACCAGA-3′

Note: NM, RNA sequence number in the NCBI Nucleotide Database (http://www.ncbi.nlm.nih.gov/nuccore); F, direct primer; R, reverse primer.

**Table 2 cells-09-01680-t002:** The content and expression of NF-kB p50, NF-kB p65, HIF-1 and HIF-2 in patients with localized and disseminated kidney cancer, as well as in metastatic sites.

Indicator	Normal Renal Parenchyma	Localized Kidney Cancer Tissue	Disseminated Kidney Cancer Tissue	RCC Metastatic Tissue
Content of NF-κB p50, NF-κB p65, HIF-1, and HIF-2				
NF-κB p50, RLU/mg of protein	4.16 (3.76–4.66)	7.6 (5.5–15.0) *	6.7 (4.8–25.4) *	6.55 (4.7–12.4) *
	Kruskal–Wallis Test: p = 0.0234			
NF-κB p65, RLU/mg of protein	4.88 (4.66–5.21)	7.5 (4.0–13.0) *	6.9 (6.3–21.5) *	3.3 (2.45–5.2) **, ***
	Kruskal–Wallis Test: p = 0.0019			
HIF-1α, RlLU/mg of protein	0.98 (0.62–1.18)	4.7 (2.2–7.8) *	5.4 (2.9–11.2) *	1.75 (1.55–2.45) *, **, ***
Kruskal–Wallis Test: p = 0.0001				
HIF-2, pg/mg of protein	248.9 (155.9–464.0)	364.5 (242.7–1308.0) *	334.9 (296.8–396.5)	1250.1 (860.8–1750.4) *, **, ***
	Kruskal–Wallis Test: p = 0.0984			
mRNA level of NF-κB p50, NF-κB p65, HIF-1, and HIF-2				
NF-κB p50, Relative Units	Used In Calculation	0.13 (0.08; 6.36)	2.00 (0.06; 13.55)	6.00 (2.05; 28.14)
	Kruskal–Wallis Test: p = 0.3289			
NF-κB p65, Relative Units	Used In Calculation	1.29 (0.08; 2.00)	1.00 (0.26; 8.00)	4.11 (0.14; 16.24)
	Kruskal–Wallis Test: p = 0.6452			
HIF-1, Relative Units	Used In Calculation	1.60 (0.50; 4.00)	0.50 (0.50; 1.80)	20.80 (9.60; 35.30) **,***
	Kruskal–Wallis Test: p = 0.0202			
HIF-2, Relative Units	Used In Calculation	1.0 (0.50; 4.00)	1.00 (0.06; 6.60)	2.95 (1.24; 12.27)
	Kruskal–Wallis Test: p = 0.8958			

Note: Ct of target gene was used in calculation (ΔΔCt method); * significant differences compared with normal renal parenchyma, *p* < 0.05; ** significant differences compared with localized kidney cancer, *p* < 0.05; *** significant differences compared with disseminated kidney cancer, *p* < 0.05.

**Table 3 cells-09-01680-t003:** The content and mRNA level of VEGF, VEGFR2 receptor and CAIX in a kidney tumor in patients with localized and disseminated kidney cancer, as well as in metastatic sites.

Indicator	Normal Renal Parenchyma	Localized Kidney Cancer Tissue	Disseminated Kidney Cancer Tissue	RCC Metastatic Tissue
Content of mRNA level of VEGF, VEGFR2, and CAIX				
VEGF, pg/mg of protein	10.28 (8.46; 14.3)	54.1 (19.7; 87.9) *	68.9 (11.1; 220.0) *	58.8 (44.0; 87.8) * ***
	Kruskal–Wallis test: p = 0.0001			
VEGFR2, pg/mg of protein	39.64 (22.3; 48.8)	35.0 (20.0; 53.0)	62.5 (28.2; 74.3) *, **	90.5 (35.6; 110.5) *, ***
	Kruskal–Wallis test: p = 1.000			
CAIX, pg/mg of protein	246.9 (111.1; 523.6)	292.2 (223.8; 1525.4)	189.3 (100.0; 199.2) **	62.4 (30.5; 78.8) *, **, ***
	Kruskal–Wallis test: p = 0.0349			
mRNA level of VEGF, VEGFR2, and CAIX				
VEGF, Relative Units	used in calculation	0.63 (0.02; 15.80)	1.03 (0.13; 42.20)	21.75 (0.50; 315.31)
	Kruskal–Wallis test: p = 0.3571			
VEGFR2, Relative Units	used in calculation	0.50 (0.13; 1.07)	1.27 (0.32; 16.00)	0.91 (0.45; 1.82) ***
	Kruskal–Wallis test: p = 0.4724			
CAIX, Relative Units	used in calculation	1.21 (0.04; 3.41)	0.04 (0.01; 1.32)	1.98 (1.14; 2.28) ***
	Kruskal–Wallis test: p = 0.3568			

Note: Ct of target gene was used in calculation (ΔΔCt method); * significant differences compared with normal renal parenchyma, *p* < 0.05; ** significant differences compared with localized kidney cancer, *p* < 0.05; *** significant differences compared with disseminated kidney cancer, *p* < 0.05.

**Table 4 cells-09-01680-t004:** The content and mRNA level of AKT, phospho-PDK1, phospho-c-RAF, phospho-GSK-3beta, phospho-PTEN, m-TOR, phospho-p70 S6 kinase and phospho-4E-BP1 in patients with localized and disseminated kidney cancer, as well as in metastatic sites.

Indicator	Localized Kidney Cancer Tissue	Disseminated Kidney Cancer Tissue	RCC Metastatic Tissue
The content of AKT, phospho-PDK1, phospho-c-RAF, phospho-GSK-3beta and phospho-PTEN, m-TOR, phospho-p70 S6 kinase and phospho-4E-BP1			
phospho -PTEN, % to normal kidney parenchyma	99.7 (70.4–139.2)	55.3 (55.2–57.0) *	50,3 (47.4–194.4) *
	Kruskal–Wallis test: p = 0.802		
AKT (total), % to normal kidney parenchyma	139.95 (110.68–229.4)	160.35 (112.3–175.4)	168.4 (113.7–224.1)
	Kruskal–Wallis test: p = 0.4802		
phospho -AKT (T308), % to normal kidney parenchyma	105.1 (76.1–156.58)	73.1 (55.9–84.7) *	64.1 (43.1–210.9) *
	Kruskal–Wallis test: p = 0.03802		
phospho -AKT (S473), % to normal kidney parenchyma	118.6 (93.2–164.1)	85.22 (53.44–106.7) *	44.1 (20.1–110.9) *
	Kruskal–Wallis test: p = 0.00802		
phospho -GSK-3β, % to normal kidney parenchyma	156.1 (106.85–250.5)	147.5 (97.4–196.9)	151.5 (100.1–227.0)
	Kruskal–Wallis test: p = 0.4802		
phospho –PDK1, % to normal kidney parenchyma	132.0 (102.35–168.0)	119.2 (111.7–132.6)	150.9 (54.2–170.2)
	Kruskal–Wallis test: p = 0.1802		
phospho -c-Raf, % to normal kidney parenchyma	157.1 (102.9–240.75)	146.35 (75.8–164.9)	164.0 (18.1–234.5)
	Kruskal–Wallis test: p = 0.5802		
m-TOR, % to normal kidney parenchyma	158.6 (90.35–218.0)	129.4 (78.8–138.4)	124.2 (50.7–194.1)
	Kruskal–Wallis test: p = 0.2802		
phospho -mTOR (Ser2448), % to normal kidney parenchyma	128.1 (93.0–205.6)	108.1 (81.7–160.4)	188.8 (72.6–201.7) *
	Kruskal–Wallis test: p = 0.3802		
phospho-p70 S6 kinase, % to normal kidney parenchyma	93.6 (67.1–117.8)	101.9 (39.0–173.7)	102.6 (31.2–195.5)
	Kruskal–Wallis test: p = 0.1641		
phospho-4E-BP1, % to normal kidney parenchyma	131.0 (78.1–188.7)	128.0 (105.4–136.8)	266.7 (105.5–489.9)
	Kruskal–Wallis test: p = 0.5284		
mRNA level of AKT, phospho-PDK1, phospho-c-Raf, phospho-GSK-3beta and phospho-PTEN, m-TOR, 70 S6 kinase and 4E-BP1			
PTEN, Relative units	0.80 (0.25; 4.72)	2.00 (1.10; 8.00) **	128.58 (0.25;402.33) ***
	Kruskal–Wallis test: p = 0.3284		
AKT, Relative units	1.19 (0.48; 4.00)	4.94 (1.74; 23.92)	0.92 (0.31; 4.75)
	Kruskal–Wallis test: p = 0.1641		
GSK-3β, Relative units	2.41 (0.46; 9.50)	4.00 (0.26; 16.00)	0.20 (0.15; 4.12)
	Kruskal–Wallis test: p = 0.3427		
PDK, Relative units	13.58 (1.25; 30.50)	1.50 (0.03; 8.00) **	0.33 (0.15; 1.18)
	Kruskal–Wallis test: p = 0.0232		
c-Raf, Relative units	1.58 (0.11; 6.99)	2.25 (0.13; 8.00)	0.15 (0.02; 5.12)
	Kruskal–Wallis test: p = 0.6720		
m-TOR, Relative units	2.69 (0.46; 13.28)	1.04 (0.34; 8.00)	0.28 (0.21; 1.15) ***
	Kruskal–Wallis test: p = 0.3802		
70S 6 kinase, Relative units	1.34 (0.50; 11.45)	1.69 (0.52; 10.25)	8.14 (0.06; 72.07)
	Kruskal–Wallis test: p = 0.9620		
4E-BP1, Relative units	0.83 (0.22; 4.46)	0.83 (0.25; 7.14)	0.21 (0.09; 5.40)
	Kruskal–Wallis test: p = 0.5005		

Note: Content of proteins in normal kidney parenchyma was indicated as 100% in Western blotting results analysis; Ct of target gene was used in calculation (ΔΔCt method); ** significant differences compared with localized kidney cancer, *p* < 0.05; *** significant differences compared with disseminated kidney cancer, *p* < 0.05.

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
