# Peer review of "Molecular Protein and Expression Profile in the Primary Tumors of Clear Cell Renal Carcinoma and Metastases"

_cells, 2020, doi:10.3390/cells9071680_

Round 1

Reviewer 1 Report

The paper entitled ‘Molecular Protein and Expression Profile in the Primary Tumors of Clear Cell Renal Carcinoma and Metastases, Role of PTEN’ is interesting, dealing with an issue of great interest in the anticancer research field. The publication of works that aim to shed the light on the molecular mechanisms underlying the metastatic spreading – and therefore the concordance/discordance of proteins expression between localized and mRCC and, among mRCC, between the primary tumor and metastatic sites - should be encouraged.

Many aspects require clarification:

  • A native English speaker should review carefully the paper, and modified grammatical errors and the construction of some sentences.

I.e. Page 7, line 218: “PTEN level and expression in localized and metastatic cancers” (this sentence has no meaning without a verb);

Page 9, line 274-275: “The overactivation 274 of the VHL/HIF/VEGF/VEGFRs axis;” (this sentence has no meaning without a verb);

Page 7, line 223:  “the associations between…” (remove ‘the’);

Page 8, line 244-246: the sentence “As a result of the study…were studied” has no meaning. I suggest to remove “as a result” and replace it with “our study aimed at investigating…”;

Figure 2, Page 8, line 264: “m-TOR it is found the growth in their protein level”. Please rephrase the sentence.

Page 9, line 287, conclusions: “The AKT signaling cascade activation was found in the primary kidney tumor, both localized and disseminated cancers”. Rephrase this sentence in a more incisive way.

Page 9, line 292, conclusions “It is found the other targets for anti-cancer therapy”. I can’t understand the meaning.

  • Title: I suggest you to modify the title of your paper, removing ‘Role of PTEN’ that is not the focus of your research.

  • Introduction:
    • You can’t talk about tumor genomic heterogeneity and heterogeneous protein function without mentioning one of the cornerstone studies on this topic in RCC [Gerlinger M et al. Intratumor heterogeneity and branched evolution revealed by multiregion sequencing. N Engl J Med. 2012 Mar 8;366(10):883-892. doi: 10.1056/NEJMoa1113205]. Similarly, the molecular characterization of RCC, which highlighted the role of alterations in the PI(3)K/AKT pathway, and the association between decreased PTEN protein levels with aggressive tumor behavior [Cancer Genome Atlas Research Network. Comprehensive Molecular Characterization of Clear Cell Renal Cell Carcinoma. Nature. 2013 Jul 4;499(7456):43-9. doi: 10.1038/nature12222] has to be discussed in the introduction section.
    • Please, after clarify the acronyms RCC (renal cell carcinoma) try to always use it in the manuscript. Moreover, specify if the tumor tissue samples analyzed in your study were all ccRCC (clear cell RCC); if so, please do not forget to indicate it in the paper.

  • Materials and Methods:
    • Please try to better clarify the definition of “normal renal parenchyma, localized tumor tissue, locally advanced, disseminated kidney cancer, and metastatic tissue”. Once you did it, I suggest to use the same classification through the paper. For instance in the results section (page 4, line 153-154) you identify 5 different categories while in the table 1 they are only four (normal renal parenchyma, tumor tissue of localized RCC, disseminated RCC, and metastatic tissue).
    • Line 82-83: as inclusion criteria you stated “histopathologically verified RCC”, it means that you enrolled RCC regardless the subtype (clear cell and non-clear cell RCC)? Please specify it in the text.
    • Line 82, 90, and 92-95: “A total of 62 patients with RCC were enrolled in the study”; “The median age of the patients was 57 years. […] Localized RCC (T1-3N0M0) was diagnosed in 18 patients, metastatic RCC (T2-4N0-1M1) in 44 patients. All patients with localized RCC underwent surgery (partial nephrectomy or simple nephrectomy) and then followed-up according to a program including regular clinical and radiological examinations.” These data should be part of the results; please remove these sentences from the Materials and Methods section and put them in the results section.
    • Line 95-96: You stated that “Patients with metastatic RCC received 2 cycles of preoperative targeted therapy with pazopanib at a dose of 800 mg daily, for 2 months”. Please specify if the biomarkers analysis was performed on pre-treated tumor tissue (deriving from the diagnostic biopsy of the primitive tumor and/or metastatic tissues). If not, this could be an important bias (analyzing tumor tissue treated with VEGFR-TKIs). I think it’s of utmost importance to specify when the analyzed tissues have been collected; if you compare the gene expression/protein levels of a biomarker between the primary tumor and the metastasis of the same patient it’s important to collect the tissues at the same time-point (also because the anti-VEGFR pazopanib inhibits exactly the angiogenic pathway that you investigated). Analogously if you want to perform a comparison with normal renal parenchyma and localized or metastatic RCC.

  • Results section:
    • I suggest adding a table (which could be even part of supplemental materials) with the main patients’ characteristics. In particular, for those patients with metastatic RCC (mRCC) it would be interesting knowing the IMDC prognostic risk group and whether the disease was metastatic at diagnosis or not (and potentially if the biomarkers expression in the primary tumor changes between the two different presentations – de-novo mRCC vs. primitive progressed).
    • Page 6, line 200: “Table 3 presents data on the expression of components of the AKT/mTOR signaling pathway in the tissue of localized, locally advanced, disseminated kidney cancer, as well as in metastatic sites.”

Again, please use a unique classification of the tumor tissue origin analyzed (locally advanced?). Then, the table 3 does not report any data about the metastatic sites (that should be added instead of the normal renal parenchyma biomarkers expression – which is not matter of comparison in this case).

  • Discussion:
    • Page 8, line 247-248: “We show the changes in protein and mRNA profiles in primary cancers and metastatic sites that are in concordance with Dagher J. et al. study [6]”. Please try to better argue/explain this sentence.
    • Page 9, line 274-285: the last paragraph of the discussion section needs to be better articulated. To date, saying that you can target the mTOR pathway as a treatment for ccRCC is reductive and obsolete. Similarly, discussion and literature on prognostic angiogenic biomarkers in RCC has been treated too superficially.

  • Conclusions:
    • I suggest you to formulate better your conclusions. Ending your paper saying that “PTEN loss […] determines the overall ccRCC patient’s survival” is misleading and improper (you have not made any correlation with patients’ survival and therefore with the patients’ prognosis).

Author Response

Dear, Reviewer,

I would like to express my gratitude for the efforts in improving the paper.

I.e. Page 7, line 218: “PTEN level and expression in localized and metastatic cancers” (this sentence has no meaning without a verb);

The correction is done

Page 9, line 274-275: “The overactivation 274 of the VHL/HIF/VEGF/VEGFRs axis;” (this sentence has no meaning without a verb);

The correction is done

Page 7, line 223:  “the associations between…” (remove ‘the’);

The correction is done

Page 8, line 244-246: the sentence “As a result of the study…were studied” has no meaning. I suggest to remove “as a result” and replace it with “our study aimed at investigating…”;

The correction is done

Figure 2, Page 8, line 264: “m-TOR it is found the growth in their protein level”. Please rephrase the sentence.

The correction is done

Page 9, line 287, conclusions: “The AKT signaling cascade activation was found in the primary kidney tumor, both localized and disseminated cancers”. Rephrase this sentence in a more incisive way.

The correction is done

Page 9, line 292, conclusions “It is found the other targets for anti-cancer therapy”. I can’t understand the meaning.

 The sentence is unclear, we have removed it.

  • Title: I suggest you to modify the title of your paper, removing ‘Role of PTEN’that is not the focus of your research.

The title is changed

  • Introduction:
    • You can’t talk about tumor genomic heterogeneity and heterogeneous protein function without mentioning one of the cornerstone studies on this topic in RCC [Gerlinger M et al. Intratumor heterogeneity and branched evolution revealed by multiregion sequencing. N Engl J Med. 2012 Mar 8;366(10):883-892. doi: 10.1056/NEJMoa1113205]. Similarly, the molecular characterization of RCC, which highlighted the role of alterations in the PI(3)K/AKT pathway, and the association between decreased PTEN protein levels with aggressive tumor behavior [Cancer Genome Atlas Research Network. Comprehensive Molecular Characterization of Clear Cell Renal Cell Carcinoma. Nature. 2013 Jul 4;499(7456):43-9. doi: 10.1038/nature12222] has to be discussed in the introduction section.

Introduction of the paper was modified. The missed data were included and discussed.

  • Please, after clarify the acronyms RCC (renal cell carcinoma) try to always use it in the manuscript. Moreover, specify if the tumor tissue samples analyzed in your study were all ccRCC (clear cell RCC); if so, please do not forget to indicate it in the paper.

We use the acronymus ccRCC. The changes were made.

  • Materials and Methods:
    • Please try to better clarify the definition of “normal renal parenchyma, localized tumor tissue, locally advanced, disseminated kidney cancer, and metastatic tissue”. Once you did it, I suggest to use the same classification through the paper. For instance in the results section (page 4, line 153-154) you identify 5 different categories while in the table 1 they are only four (normal renal parenchyma, tumor tissue of localized RCC, disseminated RCC, and metastatic tissue).

We had 4 types of materials: normal renal parenchyma, tumor tissue of localized RCC, disseminated RCC, and metastatic tissue. The correction is made.

  • Line 82-83: as inclusion criteria you stated “histopathologically verified RCC”, it means that you enrolled RCC regardless the subtype (clear cell and non-clear cell RCC)? Please specify it in the text.

The patients had only ccRCC. The correction is made.

  • Line 82, 90, and 92-95: “A total of 62 patients with RCC were enrolled in the study”; “The median age of the patients was 57 years. […] Localized RCC (T1-3N0M0) was diagnosed in 18 patients, metastatic RCC (T2-4N0-1M1) in 44 patients. All patients with localized RCC underwent surgery (partial nephrectomy or simple nephrectomy) and then followed-up according to a program including regular clinical and radiological examinations.” These data should be part of the results; please remove these sentences from the Materials and Methods section and put them in the results section.

The did not discuss about the result of the treatment. We couldn’t transfer the characteristics of patients into the results section of the paper

  • Line 95-96: You stated that “Patients with metastatic RCC received 2 cycles of preoperative targeted therapy with pazopanib at a dose of 800 mg daily, for 2 months”. Please specify if the biomarkers analysis was performed on pre-treated tumor tissue (deriving from the diagnostic biopsy of the primitive tumor and/or metastatic tissues). If not, this could be an important bias (analyzing tumor tissue treated with VEGFR-TKIs). I think it’s of utmost importance to specify when the analyzed tissues have been collected; if you compare the gene expression/protein levels of a biomarker between the primary tumor and the metastasis of the same patient it’s important to collect the tissues at the same time-point (also because the anti-VEGFR pazopanib inhibits exactly the angiogenic pathway that you investigated). Analogously if you want to perform a comparison with normal renal parenchyma and localized or metastatic RCC.

We made all investigation before the treatment. All procedures were made in biopsy samples. The effect of the targeted anti-angiogenic therapy is not an aim of the paper. the another reason is limitation in paper size. Taking into the account the mentioned above reasons, we couldn’t include this part of our investigation in the paper.

  • Results section:
    • I suggest adding a table (which could be even part of supplemental materials) with the main patients’ characteristics. In particular, for those patients with metastatic RCC (mRCC) it would be interesting knowing the IMDC prognostic risk group and whether the disease was metastatic at diagnosis or not (and potentially if the biomarkers expression in the primary tumor changes between the two different presentations – de-novo mRCC vs. primitive progressed).

The patients with metastatic ccRCC had this diagnosis before the start of the therapy. We had no progressed  cancers

  • Page 6, line 200: “Table 3 presents data on the expression of components of the AKT/mTOR signaling pathway in the tissue of localized, disseminated kidney cancer, as well as in metastatic sites.”

The corrections were made in the table. We had only the tissue of localized, disseminated kidney cancer, and metastatic sites. The data on protein content expression level are impossible, because they are used in the calculations. Content of proteins in normal kidney parenchyma was indicated as 100% in Western Blotting results analysis; Ct of target gene was used in calculation (ΔΔCt method);

Again, please use a unique classification of the tumor tissue origin analyzed (locally advanced?). Then, the table 3 does not report any data about the metastatic sites (that should be added instead of the normal renal parenchyma biomarkers expression – which is not matter of comparison in this case).

 We had 4 types of materials: normal renal parenchyma, tumor tissue of localized RCC, disseminated RCC, and metastatic tissue. The correction is made.

  • Discussion:
    • Page 8, line 247-248: “We show the changes in protein and mRNA profiles in primary cancers and metastatic sites that are in concordance with Dagher J. et al. study [6]”. Please try to better argue/explain this sentence.

The correction is made

“We show the changes in protein and mRNA profiles in primary cancers and metastatic sites. It is known the primary tumors and their corresponding metastases differ from each other in gene, mRNAs and proteins profiles [6]… “

  • Page 9, line 274-285: the last paragraph of the discussion section needs to be better articulated. To date, saying that you can target the mTOR pathway as a treatment for ccRCC is reductive and obsolete. Similarly, discussion and literature on prognostic angiogenic biomarkers in RCC has been treated too superficially.

The correction is made

“The revealed data on the significant increase in VEGFR2, m-TOR protein level and decrease in HIF-1, PTEN in metastatic sites show the promising approach in the management of disseminated ccRCC. The PTEN deficiency is a key event in the angiogenic imbalance. It results in the switching on the angiogenesis, and causes the ccRCC progression [23]. PTEN deficiency contributes to the development and progression of ccRCC. The shorter overall survival in ccRCC patients is correlated with low PTEN expression [24, 25] “

  • Conclusions:
    • I suggest you to formulate better your conclusions. Ending your paper saying that “PTEN loss […] determines the overall ccRCC patient’s survival” is misleading and improper (you have not made any correlation with patients’ survival and therefore with the patients’ prognosis).
  • The correction is made

“PTEN loss results in the downstream activation of AKT/mTOR signaling in secondary cancer lesions and determines the overall ccRCC patient’s survival. The PTEN content and mRNA level are correlated with a total AKT, GSK-3β, the 70S 6 kinases, and AKT expression. The data highlight the variety of molecular factors implicated in cancer progression and switching on the metastatic phenotype. Overactivation of angiogenic factors governing by HIF-1 leads to the increase in VEGFR2 content in metastatic sites, accompanied by the growth in m-TOR protein’s level. The AKT signaling cascade activation was found in the primary kidney tumors. NF-κB p 50, κB p 65, HIF-1, HIF-2, VEGF, CAIX, VEGFR2 mTOR, PTEN, AKT are molecular factors characterizing the distant metastases. Biological mechanisms causing particular characteristics of sites of secondary lesions serves as the predictive factors of response to anti-cancer agents. “

Reviewer 2 Report

In their publication, the authors present the differences between primary renal cell carcinomas and their metastases. They examine the expression profiles and relevant proteins. Unfortunately, it is not entirely clear how homogeneous the examined patients were. Is there any information on age and sex? The patients were pretreated. Could that have affected the results? The authors jump back and forth between RCC and ccRCC. What is the background of these different abbreviations? The methods are clearly described, but it is not clear whether linear or logarithmic scaling was used as the basis for expression levels. In some cases the differences are only very slight. The primers used are shown as continuous text, a table with a non-proportional font would be advantageous here. The investigations show that PTEN plays a central role. Figure 2 is unfortunately very difficult to understand. What do the text blocks at the top left and bottom right refer to? Unfortunately the results remain somewhat disordered.

Author Response

Dear, Reviewer,

I would like to express my gratitude for the efforts in improving the paper.

We had a group of patients with ccRCC. The expression level of target gene was calculated with help of ΔΔCt method. It is mentioned in the methods of the paper. The fold changes were calculated by ΔΔCt method (the total ΔΔCt = fold of cancerous/normal tissue gene level), using normal tissue. A ratio of specific mRNA/ GADPH (GADPH as a respective control) amplification was then calculated.

The table of primers is made.

The correction in the Material section were made. The groups of patients were homogenous with prevalence of male.

“A total of 62 patients with ccRCC were enrolled in the study (22 females and 40 males). The median age of the patients was 57.0±9.2 years.”

The description of Figure 2 was modified.

“Note: Arrows indicate changes in the studied factors. We have found a decrease in HIF-1, CAIX, PTEN, phospho-AKT content, and RNA levels of VEGFR2, m-TOR in metastases compared to the primary tumors. The NF-κB p50, NF-κB p65, HIF-2, VEGF VEGFR2, m-TOR protein’s content, and HIF-1, CAIX, PTEN, PDK RNA’s level enhance in secondary cancer lesions. Indicators with change in opposite direction are highlighted in bold. Proteins level and mRNA alterations show the specific trend in biological features of primary tumors and metastases. Despite the reduction in mRNA level of VEGFR2, m-TOR, the protein content of these molecular factors grows. In contrast, the HIF-1 and PTEN protein level shows the opposite trend. The kidney cancer aggressiveness is associated with the overactivation of AKT/mTOR signaling pathway and excess of tyrosine kinase receptors.”

Round 2

Reviewer 1 Report

The authors revised the paper according to the requested indications. The paper can be considered acceptable for publication.

Reviewer 2 Report

The authors followed well the reviewer's comments. The manuscript is now well structured and message is presented clearly.